# Detection and Risk Assessments of Multi-Pesticides in Traditional Chinese Medicine Chuanxiong Rhizoma by LC/MS-MS and GC/MS-MS

**DOI:** 10.3390/molecules27030622

**Published:** 2022-01-19

**Authors:** Delin Zhang, Yan Gou, Xingyu Yu, Mei Wang, Wen Yu, Juan Zhou, Wei Liu, Min Li

**Affiliations:** 1State Key Laboratory of Southwest Chinese Medicine Resources, Chengdu University of Traditional Chinese Medicine, Chengdu 611137, China; zhangdelintcm@126.com (D.Z.); gouyan0101@hotmail.com (Y.G.); cqqjyxy43@126.com (X.Y.); Wangmei7819@163.com (M.W.); 15208452542@163.com (W.Y.); lwiu@163.com (W.L.); 2Sichuan Institute for Drug Control, Chengdu 611731, China; zhoujuan009@163.com

**Keywords:** Chuanxiong Rhizoma, traditional Chinese medicine, pesticide residues, health risk assessment, hazard quotients, hazard index, maximum residue levels

## Abstract

With the internationalization of traditional Chinese medicines (TCMs) and the increasing use of herbal medicines around the world, there are concerns over their safety. In recent years, there have been some sporadic reports of pesticide residues in Chuanxiong Rhizoma (CX), although the lack of systematic and comprehensive analyses of pesticide residues and evaluations of toxicological risks in human health has increased the uncertainty of the potential effects of pesticides exposure in humans. This study aimed to clarify the status of pesticide residues and to determine the health risks of pesticide residues in CX. The findings of this study revealed that 99 batches of CX samples contained pesticide residues ranging from 0.05 to 3013.17 μg/kg. Here, 6–22 kinds of pesticides were detected in each sample. Prometryn, carbendazim, dimethomorph, chlorpyrifos, chlorantraniliprole, pyraclostrobin, and paclobutrazol were the most frequently detected pesticides, with detection rates of 68.69–100%. Insecticides and fungicides accounted for 43.23% and 37.84% of the total pesticides detected, respectively. Here, 86.87% of the pesticide content levels were lower than 50 μg/kg, and a small number of samples contained carbofuran, dimethoate, and isofenphos-methyl exceeding the maximum residue levels (MRLs). A risk assessment based on the hazard quotient/hazard index (HQ/HI) approach revealed that the short-term, long-term, and cumulative risks of pesticide residues in CX are well below the levels that may pose a health risk. Worryingly, six banned pesticides (carbofuran, phorate sulfone, phorate-sulfoxide, isofenphos-methyl, terbufos-sulfone, and terbufoxon sulfoxide) were detected. This study has improved our understanding of the potential exposure risk of pesticide multi-residues in CX. The results of the study will have a positive impact on improving the quality and safety of CX and the development of MRLs for pesticide residues.

## 1. Introduction

Chuanxiong Rhizoma (CX), the dried rhizome of *Ligusticum chuanxiong* Hort. (Umbelliferae), is one of the oldest and most popular herbal medicines in the world [1,2,3]. CX is widely used in clinical medicine, recorded in the “Chinese pharmacopeia” (CP, 2020 version) [1] under the Chinese name “Chuan Xiong”. More than 250 (15.81%) prescriptions containing CX are used to treat invigorating blood circulation, ischemic disorders, headache, and menstrual symptoms. Additionally, CX is also widely used in health foods, cosmetics, feed additives, spice additives, and natural preservatives [4]. Moreover, its tender leaves and stems are frequently consumed as edible food materials, such as in tossed salads, as fried vegetables, and in stews [5]. In addition to domestic demand, CX is also exported in large quantities. According to the 2017 Market Analysis report on the Circulation of Traditional Chinese Medicinal Materials released by the Ministry of Commerce of China, CX ranks 15th in in export of traditional Chinese medicines (TCMs), with an export volume of 4603.57 tons (http://sczxs.mofcom.gov.cn/article/gzdongtai/m/201806/20180602759408.shtml (accessed on 1 October 2021)). CX is mainly distributed in Sichuan Province, China, with the largest production rates in Pengzhou, Meishan, and Shifang. More than 90% of CX is produced in Sichuan, and all of the CX is planted artificially [6]. In recent years, the commercialization of CX has increased rapidly due to its large-scale use in various sectors. However, CX is vulnerable to many pests, fungal diseases, and weeds, such as root rot, powdery mildew, red spider mite, burrows, grubs, chickweed, and *Alopecurus aequalis* [7,8,9,10,11,12]. To increase the yield, reduce the cost of manual weeding, and decrease the occurrence of pests and diseases, farmers usually apply pesticides for prevention and control. Therefore, CX is prone to contamination by pesticide residues.

Pesticide residues have been found in CX and other herbal medicines in previous studies. Yi et al. reported that dichlorodiphenyltrichloroethanes (DDTs), including *o*,*p*′-DDT, *p*,*p*′-DDT, and *p*,*p*′-DDD, were detected in CX [13]. We reported that chlorpyrifos, procymidone, azoxystrobin, pyrimethanil, triazophos, and profenofos were identified in CX [14]. Furthermore, pesticide residues such as fipronil [15], dichlofluanid, trifluralin, trans-permethrin, fenpropathrin, and parathion [16] were detected in CX. It has been found that not only conventional pesticides (procymidone, azoxystrobin, pyrimethanil, etc.), but also banned pesticides (DDT, parathion, fipronil, etc.) were detected in CX. Due to the subacute and chronic toxicity of pesticide residues, people may suffer health problems from intake. Thus, controlling and regulating the use of pesticides and monitoring their levels in CX is of great concern for consumer health. China has established maximum residue limits (MRLs) for regulating the use of over 500 pesticides in foodstuffs (GB2763-2021) [17]. Regarding herbal medicines, however, equivalent MRLs are only specified for 33 pesticides (CP, 2020 version) [1]. Many studies monitoring pesticides have been conducted in China to assess the exposure risk of humans to pesticide residues in vegetables, fruits, and crops [18]. So far, there are only some reports on the establishment of detection methods or the determination of pesticide residues in CX, with low sample sizes and small production areas. Although there is a current evidence gap regarding health risk assessments of pesticide residues in CX, this is an important step that could impact public health. Therefore, it is important and urgent to systematically and comprehensively study and quantify different pesticide residues in CX and to calculate the associated risks.

It is challenging to quantify residual pesticides in CX at trace levels due to the complex nature of the matrices, such as the essential oils, Z-ligustilide, E-ligustilide, senkyunolide A, 3-butylidenephthalide, and senkyunolide O [5]. Anastassiades et al. developed an all-purpose pretreatment method for multi-residue pesticide analysis in foodstuffs, named the QuEChERS method (quick, easy, cheap, effective, rugged, and safe) [19]. So far, the QuEChERS method has been modified and applied in the pretreatment of different environmental and food samples, as well as herbal medicines, such as Chuanxiong Rhizoma, honeysuckle, Corydalis Rhizoma, Angelicae sinensis radix [14,15,16,18,20]. Tandem mass spectrometry has become prevalent because of its high degree of assurance in identification and quantification in multi-residue pesticide analysis. We have completed studies on weakly polar, volatile, and thermally stable pesticides such as organochlorine and pyrethroid in CX using gas chromatography–tandem mass spectrometry (GC-MS/MS) [14]. Ultra-performance liquid chromatography–tandem mass spectrometry (UPLC-MS/MS) was chosen to detect strongly polar, highly relative molecular masses and non-volatile pesticides in Pinelliae Rhizoma, such as organophosphates, carbamates, phenyl pyrazole derivatives, and sulfonylurea pesticides [21].

In order to understand the use of pesticides in CX, we conducted field investigations on five main producing areas, including Pengzhou, Meishan, Dujiangyan, Shifang, and Pengshan. The results showed that pesticides in the production of CX were mainly used in four areas: seed soaking, weeding, growth regulation, and pest control. For example, carbendazim and thiophanate-methyl were used for seed soaking before planting, prometryn and glyphosate were used for weeding, paclobutrazol and choline chloride were used to regulate growth, and chlorpyrifos and thiophanate-methyl were used for disease and pest control [14].

We referred to the 33 banned pesticides in the Chinese Pharmacopoeia (CP, 2020 version) [1] and the list of banned and restricted pesticides in the Ministry of Agriculture of China (http://www.zzys.moa.gov.cn/gzdt/201911/t20191129_6332604.htm (accessed on 1 October 2021)). Then, combined with the pesticides that may be involved in the production of CX and the pesticides that are more frequently detected in other herbal medicines, 136 pesticides were selected for the detection index. In this study, 99 samples were collected from main production areas (Pengzhou, Meishan, Shifang, Dujiangyan, Pengshan) and markets over three consecutive years, and the pesticide residues in CX were comprehensively analyzed. All pesticide residues detected in CX were analyzed using methods [14,21] established by our research group in the early stage. The Chinese Pharmacopoeia (CP, 2020 version) [1] and the list of prohibited and restricted pesticides from the Ministry of Agriculture of China were used as the evaluation standards. The present status of pesticide residues in CX in China was systematically studied, and the exposure and potential health risks of 37 pesticide residues detected in CX were calculated using the hazard factor (HQ) and hazard index (HI) methods [18,22,23], providing a reference for the quality and safety evaluation. Compared with previous studies, our samples represented more collection sites, a larger sample size, and had stronger representativeness. At the same time, the characteristics of GC-MS/MS and UPLC-MS/MS were used to detect a variety of pesticide residues and carry out the health risk assessment, while health risk assessments of CX have not been reported.

## 2. Results and Discussion

### 2.1. Quality Assurance of Method

The calibration curve was established and the precision and accuracy of the method were investigated to ensure the stability and reliability of the method. The accuracy and precision were expressed as the recovery, precision, and relative standard deviation at different levels, respectively (Table 1). The correlation coefficient, the average recovery rates, the precision, and the relative standard deviation (RSD) values showed ranges of 0.9984–1.0000, 76–127.1%, 0.47–6.70%, and 1.0–6.7%, respectively. The limit of detection (LODs) range was estimated at 0.003–5 µg/kg and the limit of quantification (LOQ) range was 0.01–16.67 µg/kg according to the guidelines for validation and reproducibility.

### 2.2. Pesticide Residues Concentrations

The frequency rates of detected pesticides and their residue levels in CX are listed in Table 2. A total of 37 different pesticides were detected in 99 CX samples (100%). The analyzed samples contained at least 6 and up to 22 pesticides. From Figure 1a, 34.34% (34 samples) of CX samples contained 6–9 pesticides, 44.45% (44 samples) contained 10–13 pesticides, and 21.21% (21 samples) contained more than 14 pesticides, indicating that most CX samples faced a high exposure risk to pesticides in the production process. Five types of pesticides were detected in CX, which were insecticides (16 kinds), bacteriacides (14 kinds), acaricides (2 kinds), herbicides (3 kinds), and plant growth regulators (2 kinds), among which there were more insecticides and bacteriacides, accounting for 81.08% of the total detected pesticides (Figure 1b).

The detection rate of 37 pesticides was 1.01–100%. The pesticides with residue detection rates of more than 50% included fungicides (carbendazim (100%), dimethomorph (98.99%)), herbicides (prometryn (100%)), insecticides (chlorpyrfos (89.90%), chlorantraniliprole (81.82%)), and plant growth regulators (paclobutrazol (68.69%)) (Figure 2b, Table 2). In line with Mu’s findings [8], we found that chlorpyrifos (for pest control) and thiophanate-methyl (for disease control) were commonly used pesticides during the production of CX. The high detection rates of carbendazim and dimethomorph and the low detection rate of thiophanate-methyl (6.06%) indicated that farmers used carbendazim and dimethomorph to replace thiophanate-methyl in the disease control of CX. It is speculated that this may be related to the continuous use of thiophanate-methyl to produce drug resistance [24]. The detection rate of chlorpyrifos was 89.90%, which was consistent with the high detection rate of chlorpyrifos residues in 38 batches of CX (81.58%) found by Yu [14], indicating that farmers are still using chlorpyrifos for pest control in the cultivation of CX. In addition, chlorantraniliprole also had a high detection rate, indicating that farmers combined chlorantraniliprole with chlorpyrifos for pest control in the production of CX. However, prometryn is still a widely used herbicide in the production of CX. Additionally, farmers have also used plant growth regulators such as paclobutrazol and mepiquat chloride in the production of CX to increase yields and obtain higher returns. Although the use of herbicides and plant growth regulators is prohibited in the production of CX according to GAP (Good Agriculture Practices), farmers do not strictly follow GAP in the production process, which also increases the potential risk of people being exposed to pesticides by taking CX.

It is noteworthy that no banned pesticides such as pyrethroids, organochlorines, organophosphates, or DDT, which have been reported in previous studies to be more frequently used in pests and diseases of CX, were detected in the samples [7,13,15,16]. It shows that government guidance has achieved good results. Triazophos (5.05%), profenofos (3.03%), and acetazolamide (1.01%) were only detected in market samples, while our previous study found that triazophos and profenofos were only detected in market samples, indicating that CX may be contaminated by pesticides in the process of processing, storage, and transportation. Azoxystrobin (26.26%), propoxur (8.08%), acetamiprid (2.02%), isofenphos-methyl (1.01%), terbufos-sulfone (1.01%), and thiamethoxam (2.02%) were only detected in samples collected at origin, presumably due to factors such as processing, storage, and transportation, which shorten their degradation half-life [26]. Residue levels ranged from 0.05 to 3013.17 μg/kg (Figure 2b, Table 2), and detected pesticide concentrations < 50 μg/kg accounted for 86.87% (Figure 1d), indicating that the pesticide residues vary greatly among different samples, while most of the pesticide residue levels are low.

According to the list of prohibited and restricted pesticides from the Ministry of Agriculture of China, 61 pesticides, including BHC, are banned for use on Chinese herbal medicines. Of these, 6 banned pesticides, namely carbofuran, phorate sulfone, phorate-sulfoxide, isofenphos-methyl, terbufos-sulfone, and terbufoxon sulfoxide, were detected in the samples of CX. Further, 35 samples were found to contain at least 1 of the banned pesticides and 1 sample was found to contain all six pesticides, with the detection rates being carbofuran (35.35%) > phorate-sulfoxide (21.21%) > phorate sulfone (20.20%) > terbufoxon sulfoxide (11.11%) > terbufos-sulfone, isofenphos-methyl (1.01%). The Chinese Pharmacopoeia (CP, 2020 version) [1] stipulates that 33 pesticides cannot be detected (i.e., <LOQ) in herb drugs. Among these unauthorized pesticides detected in CX, carbofuran, isofenphos-methyl, terbufos (terbufos-sulfone, terbufoxon sulfoxide), and phorate (phorate sulfone, phorate-sulfoxide) are included in the list of monitoring, but chlorpyrifos and triazophos are not included. The carbofuran in 5 samples of CX, methamidophos in 15 samples, and isosalidophos in 1 sample exceeded the limit. The European Pharmacopoeia (9.0, https://www.drugfuture.com/standard/search.aspx (accessed on 1 October 2021)) stipulates that the MRL of chlorpyrifos in herbal drugs is 0.2 mg/kg. The residual range of chlorpyrifos in CX was 0.79–134.00 μg/kg, which was lower than the limit. The maximum residue level for paclobutrazol (1.78 mg/kg) exceeded the MRLs for other species (0.5 mg/kg) established by GB2763-2021 [17]. An international list of approved pesticides and their MRLs is available, but there is no limit or standard for the MRLs of CX, so it is urgent to establish and improve the relevant standards. It is worth noting that GB2763-2021 adds MRLs for certain pesticide residues for medicinal plants or individual species (ginseng, maidenhair, Panax ginseng, wolfberry, etc.) to its previous version, GB2763-2019, and this study hopes to build on its predecessors to provide data to support the refinement of the standard. In recent years, TCMs have been affected by pesticide residues, heavy metals, and certain technical barriers in international trade, and the future status is not optimistic, seriously affecting the export of traditional Chinese medicines to generate foreign exchange and to build an international reputation. Therefore, to ensure the safety of Chinese herbal medicines, in addition to good control at the source, the whole chain of processing, transportation, and storage should be well monitored.

### 2.3. Health Risk Assessment

#### 2.3.1. Long-Term Risk Assessment

According to manuals and the available literature, the median residue level for each pesticide monitored is usually used for long-term risk analysis [18,22,23]. In the present study, the median value was used rather than the average concentration because it was higher, and consequently a decision to assume a worst-case scenario was made. In addition, the Chinese Pharmacopoeia (CP, 2020 version) [1] recommends daily consumption of CX of 0.003–0.01 kg/d. For this, a median value of 0.065 kg/d for CX was used for average consumption. For long-term risk assessments, the estimated daily intake (EDI) values are notably lower than acceptable daily intake (ADI) values, indicating that the risk from pesticide exposure via CX consumption can be ignored. Table 3 shows the HQ values calculated for CX, revealing an exposure range of 6.26 × 10^−9^–1.97 × 10^−3^. The highest HQc values were obtained for phorate (phorate sulfone and phorate-sulfoxide, 0.20%), followed by triazophos (0.04%) and isofenphos-methyl (0.01%), with the other pesticide HQc being below 0.01%. As the HQc of each detected pesticide was less than 1, the chronic intake risk of each pesticide in Chuanxiong is negligible and long-term consumption is not expected to cause health problems. Researchers have systematically summarized the effects of processing methods on pesticide residue, having found the washing, peeling, and cooking are effective ways to reduce pesticide residues in crops [27]. Specifically, in the study by Oliva et al., after zucchini was washed and blanched, the diethofencarb suffered a 96% loss and trifloxystrobin an 88.7% loss [28]. Camara et al. found that the processing, cutting, washing, and drying treatments reduced the residues of imidacloprid, tebufenozide, cypermethrin, metalaxyl, tebuconazole, and azoxystrobin in lettuce [29]. Du et al. found that after washing, peeling, homogenization, simmering, and sterilization, the residual amounts of cyflumetofen in tomatoes were significantly decreased [30]. Xiao et al. reported that processing, boiling, peeling, and drying could be useful for the partial removal of chlorpyrifos, phoxim, imidacloprid, thiamethoxam, and fenpropathrin in *Paeoniae Radix* Alba, with a removal rate reaching 98% [31]. Each processing step, including ashing, steaming, carbonization, and boiling, could significantly reduce the residues of tebuconazole, prochloraz, and abamectin in Rehmannia [26]. Most TCMs are not taken directly but are processed into decoction pieces and then used after decocting or extraction. This process helps to dissipate pesticide residues and further reduces their health risks [32]. CX materials are processed into decoction pieces and then decocted or extracted for use. Processing and decocting have a great influence on pesticide residues, so these two factors were taken into consideration in the formulation of standards. There are more than 500 types of commonly used TCMs, but unfortunately there are very few reports on the effects of processing on pesticide residues, and a great deal of research is urgently needed. In addition, a large number of studies have confirmed that washing has a good removal effect on pesticide residues of agricultural products, but there is no cleaning treatment in the production of GAP of CX, so it is suggested to add cleaning links in the production of CX.

#### 2.3.2. Short-Term Risk Assessment

Seventeen pesticides, including azoxystrobin, chlorantraniliprole, and diethofencarb, could not be included in the acute exposure risk assessment because the acute reference dose (ARfD) values were deemed unnecessary or data were not available in the Joint Meeting on Pesticide Residues (JMPR) database. The ARfD values for the other 18 pesticides are listed in Table 3, along with the corresponding acute hazard quotient (HQa) values. The results showed that the HQa values for Chuanxiong were 8.34 × 10^−^^9^–1.46 × 10^−^^3^, all of which were below 100% and within the acceptable limits. The HQa values for pesticides decreased in the following order: carbofuran (0.15%) > triazophos (0.14%) = phorate (phorate sulfone and phorate-sulfoxide, 0.14%) > pyraclostrobin (0.04%) > flutriafol (0.01%) > others (<0.01%). Despite the HQa values of all pesticides being less than 1, more attention should be paid to risk assessments, particularly for carbendazim, prometryn, dimethomorph, chlorpyrifos, chlorantraniliprole, paclobutrazol, and pyraclostrobin, given their high usage and wide range of sources for exposure, as well as potential cumulative effects.

#### 2.3.3. Cumulative Risk Assessment

Our data showed that at least six pesticide residues were detected in CX samples. Although there was no chronic risk of ingestion for each pesticide when assessed independently, the human body acts as a final accumulator of chemical pollutants, which can lead to health problems. As shown in Table 3, the chronic hazard index (HIc) and acute hazard index (HIa) values were 2.56 × 10^−^^3^ and 4.86 × 10^−^^3^, respectively; all values were less than 1. Therefore, the risk of combined exposure to pesticide residues was acceptable. The contributions of pesticides are shown in Figure 1c and Figure 2c, with insecticides and bacteriacides contributing the most to the chronic and acute hazard indices. Correspondingly, insecticides and bacteriacides were detected most frequently in CX and contained more extremely, highly, and moderately hazardous pesticides than other pesticide classes (Table 2), yielding a higher risk. Among the insecticides, phorates (phorate sulfone and phorate-sulfoxide), triazophos, and isofenphos-methyl contributed 76.98%, 15.06%, and 3.20% to the HIc, respectively, while carbofuran, phorates (phorate sulfone and phorate-sulfoxide), and triazophos contributed 30.00%, 28.66%, and 29.48% to the HIa, respectively. Consequently, we must pay more attention to these pesticides in future CX monitoring work.

Although the detected rates of carbendazim, paclobutrazol, and pyraclostrobin were high, they had little effect on the exposure risk. On the contrary, some highly toxic pesticides (e.g., chlorpyrifos, carbofuran) had low residue levels but were highly likely to pose a potential risk, reflecting the fact that some farmers are still using highly toxic or prohibited pesticides. From the results, the cumulative intake of pesticides in CX will not cause health damage [18,22,23], but attention should be paid to the use of highly toxic pesticides and their residues in the soil. Thus, additional research is needed to establish the scientific criteria (e.g., MRLs and GAP values) to determine safe limits for herbal medicines.

## 3. Materials and Methods

### 3.1. Sample Collection

A total of 99 samples of CX were collected from June 2017 to May 2019, including 45 samples from Pengzhou local farmers, 41 samples from Meishan local farmers, 3 samples from Dujiangyan local farmers, 2 samples from Shifang local farmers, and 8 samples from Medicine Market. Samples were identified by Professor Min Li (College of Pharmacy, Chengdu University of TCM). The amount of each sample was arranged from 1000 to 2000 g. These samples were immediately transported to the laboratory, pulverized mechanically to homogeneous powder, sieved through a no. 65 mesh sieve (250 μm ± 9.9 μm aperture), sealed in hermetic bags, and maintained at −4 °C until analysis. All samples for pesticide analysis were assessed using either liquid chromatography–tandem mass spectrometry (LC/MS-MS) [21] or gas chromatography–tandem mass spectrometry (GC/MS-MS) [14]. The screening involved 136 kinds of pesticide residues in CX with high usage and detection rates in the production of TCMs.

### 3.2. Reagents and Chemicals

WondaPak Quick, an easy, cheap, effective, rugged, and safe (QuEChERS) sodium acetate (NaAC) extraction kit; and WondaPak QuEChERS 15 mL C18/PSA/GC-e/silica gel purification tube (Shimadzu, Japan) analysis protective agent containing D-(+)-ribo-γ-lactone and D-sorbitol were purchased from Beijing Bailingwei Technology Co., Ltd. (Beijing, China). Ultrapure water was purified using a Milli-Q system (Millipore, Billerica, MA, USA); Primary secondary amine (PSA, 40–63 μm, 60 Å) was purchased from Tianjin Bona Aijer Technology Co., LtD (Tianjin, China) HPLC-grade acetonitrile (MeCN) and acetone (CP) were purchased from Fisher Chemical, USA. Analytical pure glacial acetic acid (GAA) was purchased from Chengdu Kelong Chemical Reagent Factory. A total of 136 pesticide reference solutions (100–1000 μg/mL) and pesticide reference substances (purity > 95%) were obtained from Tianjin Alta Scientific Co., Ltd. (Tianjin, China). (A Chem Tek, Tianjin, China) and Beijing Manhattan Biotechnology Co., Ltd. (Beijing, China). (Be Pure, Beijing, China). The internal standard triphenyl phosphate (1ST20129-1000A, 1000 μg/mL) was obtained from Tianjin Alta Scientific Co., Ltd. (A Chem Tek, China).

### 3.3. Sample Preparation and Analysis

#### 3.3.1. Sample Preparation

Here, 3.0 g of pulverized CX sample was added into a 50 mL polypropylene centrifuge tube with a screw cap, then 15 mL of 1% GAA solution was added. The tube was vortexed using a vortex mixer (IKA VORTEX GENIUS 3, VG 3 S25) to ensure even mixing. After standing for 30 min, 15 mL MeCN was added and vortexed with vigorous shaking 2000 times/min for 20 min. Subsequently, samples were chilled in the fridge for 30 min, followed by the addition of 6 g of anhydrous MgSO_4_ and 1.5 g of anhydrous NaAC, immediate shaking, vertexing with vigorous shaking 2000 times/min for 10 min, and centrifugation (3–30 k, Sigma, Osterode, Niedersachsen, Germany) at 4500 RPM for 5 min.

Next, 8 mL of supernatant was added to a dispersion solid-phase extraction clean-up tube pre-filled with clean-up material (900 mg anhydrous MgSO_4_, 300 mg PSA, 300 mg C18, 300 mg silica gel, and 90 mg graphitized carbon black) and vortexed with vigorous shaking 2000 times/min for 5 min to complete purification. After centrifugation at 8000 rpm/min for 5 min, 1 mL supernatant was accurately measured for LC-MS/MS analysis. Then, taking 3 mL of supernatant, nitrogen was dried, CP was added and dissolved, then the volume was fixed to 2 mL and shaken well. Next, we took 1 mL of solution and added 0.3 mL internal standard solution for analysis, shook this well, and used it as the test sample for GC-MS/MS.

#### 3.3.2. Sample Analysis

GC/MS-MS analysis: A Shimadzu triple-quadrupole gas chromatography–tandem mass spectrometer (GCMS-TQ8040, Shimadzu, Kyoto, Japan) equipped with an autosampler (AOC-6000, CTC Analytics, Basel, Zwingen, Switzerland), and an electron ionization (EI) source was used for the analysis of samples. An elastic quartz DB17-MS capillary column (30 m × 0.25 mm i.d., 0.25 μm, Agilent, Wilmington, DE, USA) was used for GC separation. The column velocity control mode was linear velocity. Helium was used as the carrier gas at a constant flow rate of 1.3 mL/min. The oven temperature program was set as follows: initial temperature of 60 °C (1 min hold), ramped to 160 °C at 10 °C/min, then at 2 °C/min up to 230 °C, then at 15 °C/min up to 300 °C, and finally to 320 °C at 25 °C/min (5 min hold). The temperatures of the injector port, ion source, and MS interface were maintained at 240 °C, 200 °C, and 280 °C, respectively. The monitoring mode was set to multiple-reaction monitoring (MRM). The ionization energy of the electron bombardment source was 70 eV. The injection volume was 1 μL (splitless). The GC-MS/MS dynamic multiple-reaction monitoring chromatograms of the 70 pesticide standards and CX samples are illustrated in Figure 3. LabSolutions software (Shimadzu) was employed for data acquisition and analysis. The physicochemical parameters and GC/MS-MS acquisition parameters are shown in Table 4.

UPLC/MS-MS analysis: Chromatographic separation was carried out on a 1290 Infinity Ultra High-Performance Liquid Chromatography system coupled to a 6460 Triple-Quadrupole mass spectrometer (UPLC/MS-MS, Agilent, Wilmington, DE, USA) equipped with a degasser, a binary pump, and an electrospray ionization source (AJS ESI), with dynamic multiple-reaction monitoring for detection to obtain the highest response and best sensitivity. Pesticide residuals were separated in an alternative column (Waters; Milford, MA, USA) maintained at 40 °C. The injection volume was 1 μL. The chromatographic column (Waters CORTECSTM UPLC C18 column of 150 mm × 2.1 mm i.d., 1.6 μm) was utilized for LC separation at 40 °C. The injector was operated with 1 μL volume. A mobile phase consisting of eluent A (0.1% formic acid, containing 5 mmol/L ammonium formate) and eluent B (95% acetonitrile, containing 5 mmol/L ammonium formate and 0.1% formic acid) was used at a flow rate of 0.3 mL/min. The gradient elution program was as follows: 0–2 min (80% A), 2–15 min (80%→0% A), 15–17 min (0% A), 17–17.1 min (0%→80% A), 17.1–22 min (80% A). The ion source parameters were set as follows: a capillary voltage of 3.5 kV for the positive mode and 3 kV for the negative mode; a source temperature of 150 °C; a gas temperature of 325 °C; a sheath gas (argon) flow rate of 8 L/min; a sheath gas (nitrogen) flow rate of 11 L/min; a nebulizer gas of 30 psi; and a sheath gas of 350 °C. The UHPLC-MS/MS dynamic multiple-reaction monitoring chromatograms of the 66 pesticide standards and CX samples are illustrated in Figure 4. MassHunter software (Agilent) was employed for data acquisition and analysis. The physicochemical parameters and UPLC/MS-MS acquisition parameters are shown in Table 5.

### 3.4. Risk Assessments of Pesticide Contents in CX

The hazard quotient (HQ) and hazard index (HI) were applied to quantify hazards raised by pesticide residues in CX. The acute, chronic, and cumulative risks were evaluated according to the following formula [31,34,35,36]:(1)$$\mathrm{EDI}=\frac{Ci\times IRi\times ED\times EF}{BW\times AT}$$
(2)$${\mathrm{HQ}}_{c}=\frac{EDI}{ADI}$$
where EDI (mg/kg bw/day) is the estimated daily intake; Ci (mg/kg) is the median residue level in CX [18]; IRi (kg) is the average consumption of CX (0.0065 kg) [1]; ED is the exposed days over a lifetime, which is set to 20 years [36]; EF is the exposure frequency, for which the 95th percentile of annual consumption on HM was adopted here, which was set to 90 days per year according to a questionnaire on HM consumption of 20,917 people [37]; AT is the average lifetime = 365 days × 70 years; BW is the body weight of an average adult Chinese person (60 kg). The chronic hazard quotient (HQ_c_) was calculated to assess long-term dietary exposure risk and ADI is the acceptable daily intake (mg/kg bw/day) of each pesticide, as obtained from the JMPR-FAO/WHO Joint Meeting on Pesticide Residues [33] or GB 2763-2021 (Beijing, China) [17].
(3)$$\mathrm{ESTI}=\frac{Cm\times IRm\times ED\times EF}{BW\times AT}$$
(4)$${\mathrm{HQ}}_{a}=\frac{IESI}{ARfD}$$

Here, ESTI (mg/kg bw/day) is the estimated short-term intake; Cm (mg/kg) is the highest residue level of pesticides; IRm (kg) is the 97.5th percentile of a large portion, namely the maximum daily intake of CX (0.010 kg) [1]. The acute hazard quotient (HQ_a_) was calculated to assess short-term exposure risk and ARfD (mg/kg bw/day) is the acute reference dose, as obtained from JMPR-FAO/WHO Joint Meeting on Pesticide Residues [33].
(5)HI=∑HQ

HQ_a_ and HQ_c_ of each pesticide in CX were summed up to give cumulative hazard index values for CX as HIa and HIc. An HQ of <1 or HI of <1 for any pesticide is often considered acceptable, whereas an HQ or HI of >1 is not. As the HQ or HI increases, the risk does also. For HIa > 1and HIc > 1 HM, the contribution of each pesticide was calculated by (HQ_a_/HIa) × 100% and (HQ_c_/HIc) × 100% to explore which pesticide contributes the most to HM risks [18,34,35,36,38,39,40].

### 3.5. Statistic Analysis

The statistics were organized and calculated with Microsoft Office Excel 2019. Figures were plotted using the *R* software (Version 4.0.5).

## 4. Conclusions

Herein, we investigated 136 pesticide residues in 99 CX samples collected from the production areas and herbal markets for three consecutive years using LC-MS/MS and GC-MS/MS and assessed the potential health hazards. The results revealed that a total of 37 pesticide residues were detected in CX. Each sample of CX contained 6–22 pesticide residues. The overall situation regarding pesticide residues in Chuanxiong has improved considerably, as the highly toxic banned pesticides pyrethroids, DDT, and organophosphates were not detected, but other substances, including six banned pesticides carbofuran, phorate sulfone, phorate-sulfoxide, isofenphos-methyl, terbufos-sulfone, and terbufoxon sulfoxide, were still detected. Thankfully, the detection rates of the six banned pesticides were lower than 36%, indicating that only some farmers use banned pesticides during the production of Chuanxiong. The risk assessment was based on the HQ and HI methods. The short-term, long-term, and cumulative risks of pesticides in CX were all negligible. Phorate sulfone and phorate-sulfoxide played an important role in the cumulative HI.

Combined with these results, it could be concluded that the high pesticide residue concentrations in individual samples and the widespread use of highly toxic pesticides are the main causes of risk. Therefore, we make the following suggestions for future actions:Strengthen the scientific rationale for the use of pesticides, as well as for the supervision of banned and restricted pesticides;Speed up the registration of pesticides for CX via the Regulations of the People’s Republic of China on Pesticide Management, (http://www.moa.gov.cn/gk/zcfg/xzfg/201704/t20170405549362.htm (accessed on 1 October 2021)) and select high-efficiency, low-toxicity biological pesticides instead of prohibited and restricted pesticides;Establish the whole process of supervision for the production and circulation of CX planting, processing, storage, and transportation.

## Figures and Tables

**Figure 1 molecules-27-00622-f001:**
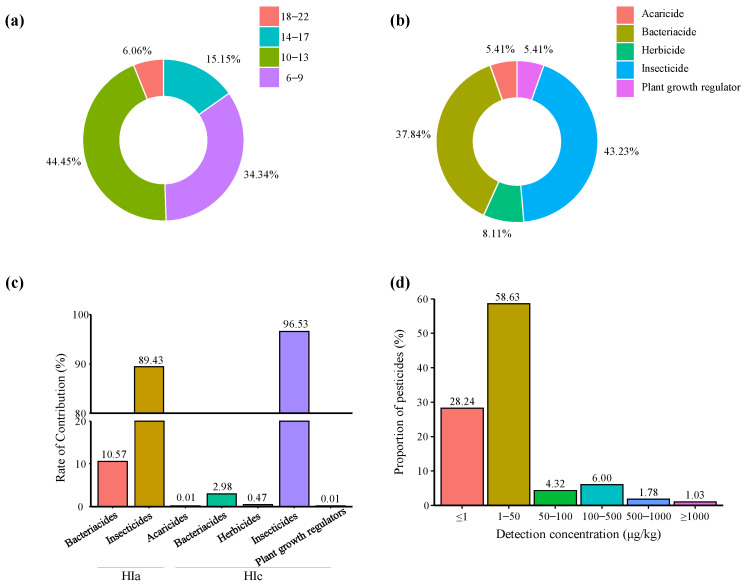
Pesticide residues in a single sample of Chuanxiong Rhizoma (**a**). The proportions of different types of pesticides detected (**b**). Contribution rates of different types of pesticides for HIa or Hic (**c**). Proportions of pesticides detected in different concentration ranges (**d**).

**Figure 2 molecules-27-00622-f002:**
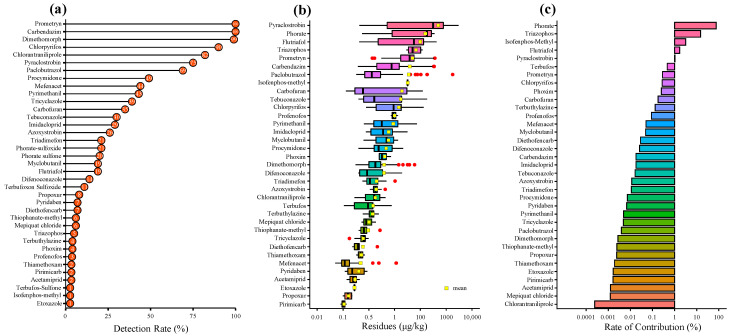
Detection rates of each pesticide (**a**). Residues of each pesticide (**b**). The HI percentages of each pesticide calculated for CX (**c**).

**Figure 3 molecules-27-00622-f003:**
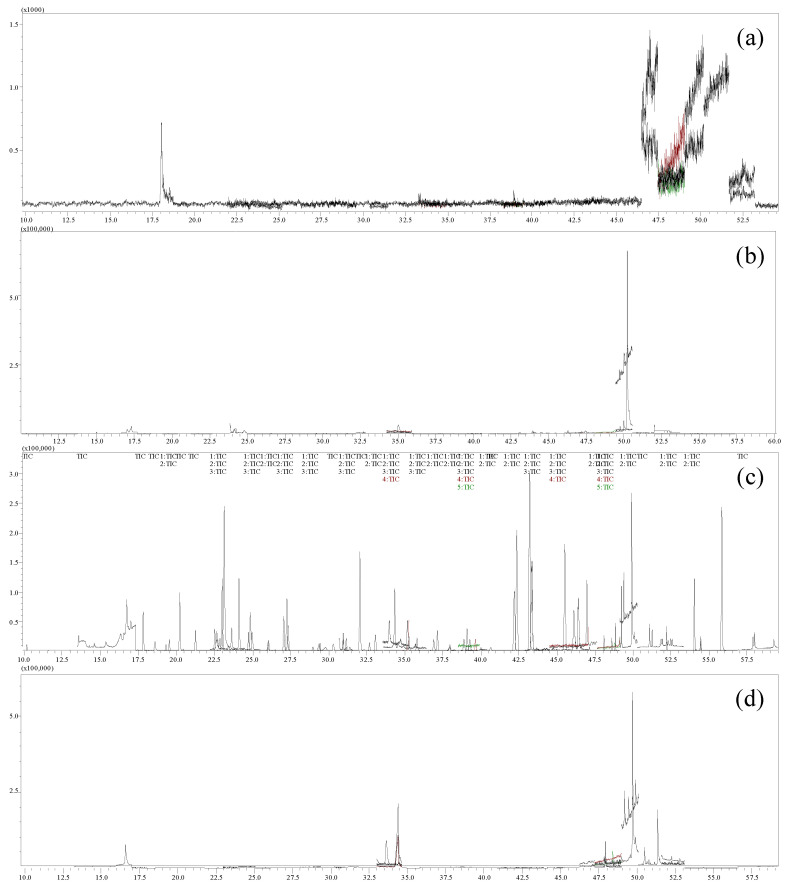
TIC for blank (**a**). TIC for recovery (**b**). TIC for mixed reference solution (**c**). TIC for sample (**d**) (GC-MS/MS).

**Figure 4 molecules-27-00622-f004:**
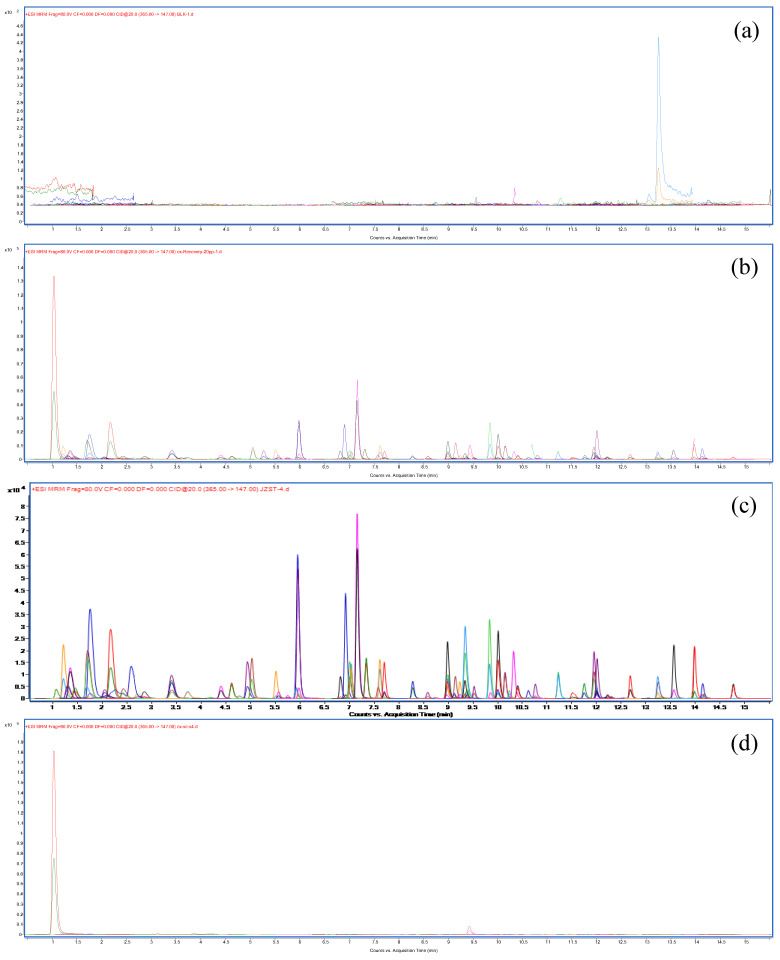
TIC for blank (**a**). TIC for recovery (**b**). TIC for mixed reference solution (**c**). TIC for sample (**d**) (LC-MS/MS).

**Table 1 molecules-27-00622-t001:** Quality assurance parameters obtained for the analysis of the target pesticides (n = 6).

No.	Pesticide *	Method	Calibration Curve	*r*	Linear Range (ng/mL)	LOD (μg/kg)	LOQ (μg/kg)	Recovery/RSD (%)	Precision/RSD (%)
1	Azoxystrobin [B]	GC-MS/MS	y = 1.6974x − 0.1234	0.9997	2.00–200.08	5.00	16.67	99.67 ± 3.33	1.78
2	Chlorpyrifos [I]		y = 4.6857x + 0.0105	0.9993	1.00–100.07	0.10	0.33	100.33 ± 6.20	1.07
3	Paclobutrazol [P]		y = 10.1504x − 0.0864	0.9997	1.00–99.99	1.00	3.33	127.10 ± 1.70	1.12
4	Procymidone [B]		y = 4.8649x − 0.0257	0.9997	1.00–100.08	0.20	0.67	80.75 ± 6.00	1.91
5	Profenofos [I]		y = 2.9991x − 0.1052	0.9997	9.98–998.40	0.30	1.00	102.67 ± 4.53	2.19
6	Pyrimethanil [B]		y = 6.8548x + 0.0241	0.9997	1.00–100.00	4.00	13.33	99.83 ± 7.00	6.70
7	Triazophos [I]		y = 3.9022x − 0.1621	0.9998	2.00–199.52	0.40	1.33	105.67 ± 5.93	2.27
8	Acetamiprid [I]	LC-MS/MS	y = 3613.5381x + 50.7307	0.9999	0.80–80.00	0.04	0.13	89.47 ± 1.12	0.47
9	Carbendazim [B]		y = 5877.5340x − 281.5931	0.9989	1.00–19.90	0.20	0.67	90.00 ± 2.17	1.18
10	Carbofuran [I]		y = 6817.9976x − 144.2714	0.9998	5.03–503.00	0.10	0.33	94.86 ± 1.09	0.79
11	Chlorantraniliprole [I]		y = 1386.3755x − 379.9283	0.9997	2.40–120.18	0.04	0.13	93.04 ± 2.31	2.96
12	Diethofencarb [B]		y = 2247.1472x − 101.1371	0.9999	0.80–39.96	0.10	0.33	91.90 ± 3.87	1.13
13	Difenoconazole [B]		y = 1457.2105x + 63.9727	0.9994	1.00–99.60	0.20	0.67	89.39 ± 1.67	1.62
14	Dimethomorph [B]		y = 3451.6049x − 276.4722	0.9997	1.00–50.22	0.10	0.33	91.20 ± 1.76	1.09
15	Etoxazole [A]		y = 10024.1485x − 74.2240	0.9998	0.24–12.00	0.03	0.10	84.03 ± 1.23	1.80
16	Flutriafol [B]		y = 2339.5870x − 51.3784	0.9998	2.39–238.51	0.10	0.33	94.88 ± 3.09	1.20
17	Imidacloprid [I]		y = 996.5774x + 107.5277	0.9984	2.01–201.12	0.10	0.33	107.07 ± 1.34	1.41
18	Isofenphos-Methyl [I]		y = 4.5960x − 0.0197	0.9997	2.00–200.00	0.20	0.67	114.46 ± 1.56	2.49
19	Mefenacet [H]		y = 11117.9465x − 758.7031	1	1.01–50.30	0.03	0.10	89.69 ± 1.33	0.75
20	Mepiquat Chloride [P]		y = 1986.2665x + 32.5621	0.9995	0.98–98.20	0.20	0.67	45.30 ± 4.73	1.01
21	Myclobutanil [B]		y = 930.0140x − 14.2133	0.9996	1.00–50.10	0.20	0.67	92.35 ± 1.60	2.87
22	Phorate Sulfone [I]		y = 435.7915x + 61.2286	0.9993	2.00–100.12	0.40	1.33	88.60 ± 2.04	2.25
23	Phorate-Sulfoxide [I]		y = 3070.6656x − 196.0316	0.9998	2.00–200.24	0.20	0.67	91.94 ± 1.57	0.76
24	Phoxim [I]		y = 339.0084x − 19.5667	0.9993	1.00–50.05	1.00	3.33	97.49 ± 5.30	3.61
25	Pirimicarb [I]		y = 8456.4438x − 279.4627	0.9999	1.00–50.00	0.02	0.07	94.17 ± 1.0	0.82
26	Prometryn [H]		y = 8271.0951x − 530.4005	0.9991	0.40–40.00	0.003	0.01	96.34 ± 1.04	1.79
27	Propoxur [I]		y = 6167.7557x − 489.1121	0.9997	1.00–100.20	0.08	0.27	90.59 ± 1.73	0.82
28	Pyraclostrobin [B]		y = 1833.6016x + 1839.3435	0.9991	0.99–99.24	0.30	1.00	86.56 ± 4.18	2.37
29	Pyridaben [A]		y = 9412.2906x + 178.4811	0.9999	0.24–12.01	0.07	0.23	86.67 ± 4.65	0.96
30	Tebuconazole [B]		y = 1340.5254x − 169.6070	0.9998	5.93–296.40	0.40	1.33	88.74 ± 1.88	1.32
31	Terbufos-Sulfone [I]		y = 1287.3285x − 87.3501	0.9998	1.00–50.16	0.20	0.67	93.79 ± 1.63	1.90
32	Terbufoxon Sulfoxide [I]		y = 9252.2302x − 600.9546	0.9996	1.01–101.18	0.20	0.67	87.62 ± 2.82	0.85
33	Terbuthylazine [H]		y = 3045.6663x − 248.6583	0.9998	0.99–49.60	0.10	0.33	88.09 ± 1.12	0.97
34	Thiamethoxam [I]		y = 2953.8961x + 134.7366	0.9994	0.99–99.43	0.10	0.33	98.77 ± 1.56	2.03
35	Thiophanate-Methyl [B]		y = 3566.9849x − 671.2516	0.9994	1.03–51.55	0.10	0.33	76.00 ± 6.40	5.02
36	Triadimefon [B]		y = 1236.5048x − 72.8577	0.9998	1.20–119.98	0.50	1.67	93.91 ± 2.46	1.57
37	Tricyclazole [B]		y = 3332.3058x − 291.4985	0.9996	1.01–100.52	0.04	0.13	84.67 ± 2.37	0.70

* [I] = Insecticide; [B] = Bacteriacide; [A] = Acaricide; [P] = Plant Growth Regulator; [H] = Herbicide.

**Table 2 molecules-27-00622-t002:** Frequencies of detected pesticides and their residue levels in Chuanxiong Rhizoma (n = 99).

No.	Pesticide	Available Range	Number of Batches Detected	Detected Rate/%	Range (μg/kg)	Median (μg/kg)	Mean ± SD (μg/kg)	MRL (mg/kg) *
1	Carbendazim	Common	99	100.00%	0.38–343.55	7.77	38.92 ± 83.68	0.02–20.00 a
2	Prometryn	Common	99	100.00%	0.31–59.48	38.49	49.56 ± 59.09	0.02–0.50 a
3	Dimethomorph	Common	98	98.99%	1.38–364.44	1.82	3.95 ± 7.94	0.01–40.00 a
4	Chlorpyrifos	Common	89	89.90%	0.79–134.00	9.22	14.69 ± 22.04	0.20 d
5	Chlorantraniliprole	Common	81	81.82%	0.28–4.49	1.64	1.80 ± 1.12	0.01–40.00 a
6	Pyraclostrobin	Common	74	74.75%	0.44–3013.17	316.96	496.39 ± 617.16	0.02–30.00 a
7	Paclobutrazol	Common	68	68.69%	0.33–1780.00	1.33	35.52 ± 216.49	0.05–0.50 a
8	Procymidone	Common	49	49.49%	0.40–21.57	2.46	4.72 ± 4.74	0.20–30 a
9	Mefenacet	Common	44	44.44%	0.05–11.64	0.12	0.49 ± 1.77	0.05 a
10	Pyrimethanil	Common	43	43.43%	0.66–74.60	3.26	8.91 ± 12.66	0.01–20.00 a
11	Tricyclazole	Common	39	39.39%	0.18–0.98	0.65	0.62 ± 0.2	0.50–5.00 a
12	Carbofuran	Forbidden	35	35.35%	0.13–124.17	0.61	19.26 ± 31.1	0.05 b
13	Tebuconazole	Common	30	30.30%	0.40–184.44	1.65	17.26 ± 35.88	0.05–40.00 a
14	Imidacloprid	Common	29	29.29%	0.78–31.90	3.61	5.91 ± 6.53	0.01–20.00 a
15	Azoxystrobin	Common	26	26.26%	1.11–4.39	1.82	1.96 ± 0.75	0.01–70.00 a
16	Phorate Sulfone and Phorate-Sulfoxide	Forbidden	21	21.21%	0.55–355.90	180.73	156.44 ± 125.09	0.02 b
17	Triadimefon	Common	21	21.21%	0.57–10.55	1.12	2.02 ± 2.26	0.01–10.00 a
18	Flutriafol	Common	19	19.19%	0.43–423.29	57.02	98.6 ± 112.48	0.01–10.00 a
19	Myclobutanil	Common	19	19.19%	0.63–13.64	4.93	5.8 ± 4.18	0.01–20.00 a
20	Difenoconazole	Common	14	14.14%	0.39–19.67	0.87	3.9 ± 6.1	0.01–10.00 a
21	terbufos	Forbidden	11	11.11%	0.11–7.76	0.94	1.49 ± 2.19	0.01–0.05 a
22	Propoxur	Common	8	8.08%	0.10–0.23	0.16	0.17 ± 0.05	0.05–1.00 c
23	Diethofencarb	Common	7	7.07%	0.23–2.16	0.39	0.6 ± 0.69	0.2–5.00 a
24	Pyridaben	Common	7	7.07%	0.12–0.88	0.23	0.41 ± 0.32	0.1–5.00 a
25	Mepiquat Chloride	Common	6	6.06%	0.53–1.89	0.81	1.03 ± 0.55	0.05–5.00 a
26	Thiophanate-Methyl	Common	6	6.06%	0.41–2.73	0.66	0.97 ± 0.88	0.1–5.00 a
27	Triazophos	Common	5	5.05%	29.00–122.00	50.50	67.78 ± 42.12	0.05–1.00 a
28	Phoxim	Common	4	4.04%	2.46–7.37	3.38	4.15 ± 2.29	0.05–0.30 a
29	Terbufos-Sulfone and Terbufoxon Sulfoxide	Common	4	4.04%	0.59–2.46	1.33	1.43 ± 0.77	0.02 b
30	Profenofos	Common	3	3.03%	7.21–13.70	9.01	9.97 ± 3.35	0.01–20.00 a
31	Acetamiprid	Common	2	2.02%	0.14–0.45	0.30	0.3 ± 0.22	0.01–10.00 a
32	Pirimicarb	Common	2	2.02%	0.08–0.14	0.11	0.11 ± 0.04	0.01–20.00 a
33	Thiamethoxam	Common	2	2.02%	0.33–0.71	0.52	0.52 ± 0.27	0.01–10.00 a
34	Etoxazole	Common	1	1.01%	0.28–0.28	0.28	0.28	0.01–15.00 a
35	Isofenphos-Methyl	Forbidden	1	1.01%	32.20–32.20	32.20	32.20	0.02 b

* a: GB2763-2021 (Beijing China) [17]; b: Ch.P (2020) [1]; c: European Commission Reg. (EC) No 149/2008 [25]; d: European Pharmacopoeia (9.0) (https://www.drugfuture.com/standard/search.aspx (accessed on 1 October 2021)).

**Table 3 molecules-27-00622-t003:** Long-term and short-term risk assessments of Chuanxiong Rhizoma.

Pesticide	EDI (mg/kg bw/day)	ADI (mg/kg bw/day) *	HQc	ESTI (mg/kg bw/day)	ARfD (mg/kg bw/day)	HQa
Acetamiprid	2.25 × 10^−9^	0.07 a	3.22 × 10^−^^8^	5.28 × 10^−^^9^	0.10 a	5.28 × 10^−^^8^
Azoxystrobin	1.39 × 10^−8^	0.20 a	6.93 × 10^−^^8^	5.15 × 10^−^^8^	-	/
Carbendazim	5.93 × 10^−8^	0.03 a	1.98 × 10^−^^6^	4.03 × 10^−^^6^	0.50 a	8.07 × 10^−^^6^
Carbofuran	4.66 × 10^−9^	0.001 a	4.66 × 10^−^^6^	1.46 × 10^−^^6^	0.001 a	1.46 × 10^−^^3^
Chlorantraniliprole	1.25 × 10^−8^	2.00 a	6.26 × 10^−^^9^	5.27 × 10^−^^8^	-	/
Chlorpyrifos	7.04 × 10^−8^	0.01 a	7.04 × 10^−^^6^	1.57 × 10^−^^6^	0.10 a	1.57 × 10^−^^5^
Diethofencarb	2.98 × 10^−9^	0.004 b	7.44 × 10^−^^7^	2.54 × 10^−^^8^	-	/
Difenoconazole	6.64 × 10^−9^	0.01 a	6.64 × 10^−^^7^	2.31 × 10^−^^7^	0.30 a	7.70 × 10^−^^7^
Dimethomorph	1.39 × 10^−8^	0.20 a	6.93 × 10^−^^8^	6.98 × 10^−^^7^	0.60 a	1.16 × 10^−^^6^
Etoxazole	2.14 × 10^−9^	0.05 a	4.27 × 10^−^^8^	3.29 × 10^−^^9^	-	/
Flutriafol	4.35 × 10^−7^	0.01 a	4.35 × 10^−^^5^	4.97 × 10^−^^6^	0.05 a	9.94 × 10^−^^5^
Imidacloprid	2.76 × 10^−8^	0.06 a	4.59 × 10^−^^7^	3.75 × 10^−^^7^	0.40 a	9.36 × 10^−^^7^
Isofenphos-Methyl	2.46 × 10^−7^	0.003 b	8.19 × 10^−^^5^	3.78 × 10^−^^7^	-	/
Mefenacet	9.16 × 10^−10^	0.0007 b	1.31 × 10^−^^6^	1.37 × 10^−^^7^	-	/
Mepiquat Chloride	6.18 × 10^−9^	0.195 b	3.17 × 10^−^^8^	2.22 × 10^−^^8^	-	/
Myclobutanil	3.76 × 10^−8^	0.03 a	1.25 × 10^−^^6^	1.60 × 10^−^^7^	-	/
Paclobutrazol	1.01 × 10^−8^	0.10 b	1.01 × 10^−^^7^	2.09 × 10^−^^5^	-	/
Phorate Sulfone and Phorate-Sulfoxide	1.38 × 10^−6^	0.0007 a	1.97 × 10^−^^3^	4.18 × 10^−^^6^	0.003 a	1.39 × 10^−^^3^
Phoxim	2.58 × 10^−8^	0.004 b	6.45 × 10^−^^6^	8.65 × 10^−^^8^	-	/
Pirimicarb	8.40 × 10^−10^	0.02 a	4.20 × 10^−^^8^	1.64 × 10^−^^9^	0.10 a	1.64 × 10^−^^8^
Procymidone	1.88 × 10^−8^	0.10 a	1.88 × 10^−^^7^	2.53 × 10^−^^7^	0.10 a	2.53 × 10^−^^6^
Profenofos	6.88 × 10^−8^	0.03 a	2.29 × 10^−^^6^	1.61 × 10^−^^7^	1.00 a	1.61 × 10^−^^7^
Prometryn	2.94 × 10^−7^	0.04 b	7.34 × 10^−^^6^	4.28 × 10^−^^6^	-	/
Propoxur	1.22 × 10^−9^	0.02 b	6.11 × 10^−^^8^	2.70 × 10^−^^9^	-	/
Pyraclostrobin	2.42 × 10^−6^	0.09 a	2.69 × 10^−^^5^	3.54 × 10^−^^5^	0.09 a	3.93 × 10^−^^4^
Pyridaben	1.76 × 10^−9^	0.01 b	1.76 × 10^−^^7^	1.03 × 10^−^^8^	-	/
Pyrimethanil	2.49 × 10^−8^	0.20 a	1.24 × 10^−^^7^	8.76 × 10^−^^7^	-	/
Tebuconazole	1.26 × 10^−8^	0.03 a	4.20 × 10^−^^7^	2.17 × 10^−^^6^	0.30 a	7.22 × 10^−^^6^
Terbufos-Sulfone and Terbufoxon Sulfoxide	7.17 × 10^−9^	0.0006 a	1.20 × 10^−^^5^	9.11 × 10^−^^8^	0.002 a	4.56 × 10^−^^5^
Terbuthylazine	1.01 × 10^−8^	0.003 b	3.37 × 10^−^^6^	2.89 × 10^−^^8^	-	/
Thiamethoxam	3.97 × 10^−9^	0.08 a	4.96 × 10^−^^8^	8.34 × 10^−^^9^	1.00 a	8.34 × 10^−^^9^
Thiophanate-Methyl	5.00 × 10^−9^	0.08 a	6.25 × 10^−^^8^	3.21 × 10^−^^8^	-	/
Triadimefon	8.55 × 10^−9^	0.03 a	2.85 × 10^−^^7^	1.24 × 10^−^^7^	0.08 a	1.55 × 10^−^^6^
Triazophos	3.85 × 10^−7^	0.001 a	3.85 × 10^−^^4^	1.43 × 10^−^^6^	0.001 a	1.43 × 10^−^^3^
Tricyclazole	4.96 × 10^−9^	0.04 b	1.24 × 10^−^^7^	1.15 × 10^−^^8^	-	/

* a: JMPR [33], b: GB2763-2021 [20] (http://www.cnhfa.org.cn/fagui/show.php?itemid=602 (accessed on 1 October 2021)); -: no related data are displayed; /: no results calculated.

**Table 4 molecules-27-00622-t004:** Physicochemical parameters and GC/MS-MS acquisition parameters.

No.	Pesticide	Retention Time (min)	Quantitative Ion Pair (*m*/*z*)	Collision Energy (eV)	Qualitative Ion Pair 1 (*m*/*z*)	Collision Energy 1 (eV)	Qualitative Ion Pair 2 (*m*/*z*)	Collision Energy 2 (eV)	ME/%
1	Azoxystrobin	59.237	344.1 > 183.1	24	344.1 > 329.1	16	344.1 > 156.1	32	93
2	Chlorpyrifos	30.055	313.9 > 257.9	14	313.9 > 285.9	8	313.9 > 193.9	28	115
3	Paclobutrazol	35.505	236.1 > 125.0	14	236.1 > 167.0	10	236.1 > 132.0	16	104
4	Procymidone	34.91	283.0 > 96.0	10	283.0 > 255.0	12	283.0 > 68.0	24	146
5	Profenofos	38.81	336.9 > 266.9	14	336.9 > 308.9	6	336.9 > 294.9	10	98
6	Pyrimethanil	23.785	198.1 > 183.1	14	198.1 > 158.1	18	198.1 > 118.1	28	105
7	Triazophos	47.985	257.0 > 162.0	8	257.0 > 134.0	22	257.0 > 119.0	26	106
8	Triphenyl phosphate (internal standard)	48.75	326.0 > 233.0	10	326.0 > 215.0	25	326.0 > 169.0	30	90

**Table 5 molecules-27-00622-t005:** Physicochemical parameters and UPLC-MS/MS acquisition parameters.

No.	Pesticide	Fragmentor (V)	Retention Time (min)	Quantitative Ion Pair (*m*/*z*)	Collision Energy (eV)	Qualitative Ion Pair (*m*/*z*)	Collision Energy (eV)	ME/%
1	acetamiprid	80	4.409	223.0 > 126.0	15	223.0 > 56.0	15	79.4
2	Carbendazim	92	1.695	192.3 > 160.1	16	192.3 > 132.1	32	93.4
3	carbofuran	90	7.155	222.1 > 165.1	5	222.1 > 123.1	17	80.5
4	chlorantraniliprole	102	8.991	484.0 > 453.0	13	484.0 > 286.0	10	120.2
5	diethofencarb	80	9.526	268.0 > 226.0	5	268.0 > 152.0	20	86.6
6	difenoconazole	160	11.509	406.0 > 251.0	20	406.0 > 337.0	15	92.7
7	Dimethomorph	120	9.236	388.0 > 301.0	20	388.0 > 165.0	25	118.8
8	etoxazole	110	14.149	360.5 > 141.1	32	360.5 > 113.1	60	117
9	flutriafol	81	7.705	302.0 > 70.2	16	302.0 > 123.1	32	85.8
10	imidacloprid	80	3.736	256.0 > 209.0	10	256.0 > 175.0	10	85.9
11	isofenphos-methyl	100	12.011	231.0 > 121.0	15	231.0 > 199.0	15	88.5
12	Mefenacet	60	10.011	299.1 > 148.1	10	299.1 > 120.1	30	82.8
13	mepiquat chloride	105	1.068	114.0 > 58.2	28	114.0 > 98.2	28	83.3
14	myclobutanil	120	9.852	289.0 > 70.0	15	289.0 > 125.0	20	97.3
15	Phorate sulfone	65	9.122	293.0 > 171.0	8	293.0 > 247.0		81.9
16	Phorate-sulfoxide	80	7.615	277.0 > 97.0	25	277.0 > 171.0	15	94.8
17	phoxim	60	12.303	299.0 > 77.2	32	299.0 > 129.1	8	96.2
18	pirimicarb	120	4.942	239.0 > 72.0	20	239.0 > 182.0	15	83.9
19	prometryn	120	9.341	242.2 > 158.1	20	242.2 > 200.2	20	87.8
20	propoxur	80	7.005	210.0 > 111.0	10	210.0 > 168.0	5	94.4
21	Pyraclostrobin	92	11.998	388.4 > 194.1	16	388.4 > 163.1	8	114.1
22	pyridaben	80	14.767	365.0 > 309.0	10	365.0 > 147.0	20	85.6
23	tebuconazole	120	10.32	308.0 > 70.0	20	308.0 > 151.0	20	83.2
24	Terbufos-Sulfone	75	10.242	321.1 > 171.2	4	321.1 > 97.0	48	94.8
25	Terbufoxon Sulfoxide	60	8.985	305.1 > 187.2	4	305.1 > 97.0	50	91.5
26	Terbuthylazine	102	9.342	230.0 > 174.1	12	230.0 > 68.2	40	88.4
27	thiamethoxam	80	2.429	292.0 > 211.0	5	292.0 > 181.0	20	81.4
28	thiophanate-methyl	105	6.825	343.1 > 151.0	17	343.1 > 117.9	65	106.7
29	triadimefon	120	10.003	294.0 > 69.0	20	294.0 > 197.0	15	85.6
30	tricyclazole	120	4.62	190.0 > 163.0	25	190.0 > 136.0	30	95.3

## Data Availability

Not applicable.

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
