# Peer review of "Detection and Risk Assessments of Multi-Pesticides in Traditional Chinese Medicine Chuanxiong Rhizoma by LC/MS-MS and GC/MS-MS"

_molecules, 2022, doi:10.3390/molecules27030622_

Round 1

Reviewer 1 Report

There is the need to include additional literature references and also a comparison of your results with other published results.  Please include a TIC for a blank, a recovery sample and a positive sample, to demonstrate the goodness of the analytical methods for both GC-MSMS and LC-MSMS. Please improve the English language and reformat the document to introduction, materials and methods, discussion of the results, conclusions.

Author Response

Dear editors and references, thank you for your very careful review and constructive comments on our manuscript (manuscript No.: molecules-1467711) " Detection and risk assessments of multi-pesticides in traditional Chinese medicine Chuanxiong Rhizoma by LC/MS-MS and GC/MS-MS ". Taking into account all the corrections and suggestions, the manuscript has been revised. We are very grateful to the reviewers for pointing out that our manuscript needs to be improved. The points raised in the reply are as follows:

Point 1: There is the need to include additional literature references and also a comparison of your results with other published results.

Response 1: According to the reviewer's opinion, we added and compared the published pesticide residue detection results in document “ 2.2.Pesticide Residues Concentrations”.

Point 2: Please include a TIC for a blank, a recovery sample, and a positive sample, to demonstrate the goodness of the analytical methods for both GC-MSMS and LC-MSMS.

Response 2: According to the reviewer's opinion, we added the TIC for a blank, a recovery sample, and a positive sample in Figure 3 and Figure 4.

Point 3: Please improve the English language and reformat the document to introduction, materials and methods, discussion of the results, conclusions.

Response 2: The English language was modified according to the opinions of reviewers. The document format is written about the “molecules-template”. The template requires the format of introduction, result, discussion, material and method, and conclusion.

Reviewer 2 Report

This is an interesting manuscript that describes a study that seeks to consider the levels and thus exposure and risk from pesticide residues present in a common herb, Chuanxiong Rhizoma (CX), used in traditional Chinese medicine. The manuscript topic does fall within the scope of the journal but I believe it would gain much more exposure in a journal more aligned with public health.

Overall the approach adopted appears sound but I am not at all convinced by the health risk assessment. The approach explained within the manuscript is highly simplistic and is not aligned with pesticide risk assessments in other sectors e.g. that in standard western medicine. The use of hazard indices are uninformative and give no real useful information as to which population grounds are at risk.  Please justify this approach and discuss its limitations. There are standard international approaches for evaluating pesticide risk.

Another main concern is with the article presentation. Use of the English language is not poor but the manuscript does contain many grammatical errors and some rather odd phraseology and word choice. For example at line 75 the text states ‘main grass harm’. I do not quite understand what this means. Grass is not being harmed. Grass  does not contaminate the crop. Similarly at line 60 the text states ‘cultivated artificially’ - what does this mean? I would strongly suggest the manuscript is proof read and corrected by a native English speaker before resubmission. Or sent to a professional editing service.

In addition the structure of the manuscript is quite poor. There is a great deal of information in the Results section which is not results but Methodology.  Paragraphing in the introduction needs a great deal of attention. The article is also far too long. There is a considerable amount of superfluous information throughout the manuscript which could easily be removed without loosing vital information. This would make the article more readable. For example, a little background is interesting in the first paragraph or two but should be removed elsewhere. Much of the background and general chat about CX in the introduction is not necessary. Elsewhere in the manuscript information given in Section 1 is repeated. Please avoid repetition. At line 298 the sentence repeats information elsewhere.

Individuals should not be named in the text e.g Line 305. Indeed most of the important information in section 3.1 could be presented in a table. Pesticide analysis information could easily be cut down. Figure 4 adds very little value.

A short summary of the methodological approach should be included in the Abstract.

The methodology section needs more work. Method description in the results section needs to repositioned. Approaches needed to be justified.

Please take care to ensure that text in the Results section does not simply repeat what is obvious from the data in tables. Important insights only should be repeated and these should add value to the tables.

Referencing is also inadequate in places. There are many stated facts that are not referenced. For example at line 152 a reference is needed related to reduced degradation half-lives and references are needed to support the manuals and literature mentioned at line 224. There are several other examples in the text.

Other comments

- All abbreviations in the main text should be spelt out the first time they are used. Abbreviations should be avoided in the abstract. SFDA needs to spelt out in full at line 50 as well as TCM at line 60.
- hyperlinks in the text should be converted to a reference.
- At line 74 ‘red spiders’ should read ‘red spider mite’. These insects are not arachnids.
- why was dichlorvos mentioned at line 104 rather than other pesticides? 

Author Response

Dear editors and references, thank you for your very careful review and constructive comments on our manuscript (manuscript No.: molecules-1467711) " Detection and risk assessments of multi-pesticides in traditional Chinese medicine Chuanxiong Rhizoma by LC/MS-MS and GC/MS-MS ". Taking into account all the corrections and suggestions, the manuscript has been revised. We are very grateful to the reviewers for pointing out that our manuscript needs to be improved. The points raised in the reply are as follows:

Point 1: Overall the approach adopted appears sound but I am not at all convinced by the health risk assessment. The approach explained within the manuscript is highly simplistic and is not aligned with pesticide risk assessments in other sectors e.g. that in standard western medicine. The use of hazard indices are uninformative and give no real useful information as to which population grounds are at risk.  Please justify this approach and discuss its limitations. There are standard international approaches for evaluating pesticide risk.

Response 1: The Hazard Quotient (HQ) and Hazard Index (HI) were applied to health risk assessments. HQ uses exposure assessment. Exposure assessment is the combination of food chemical content data and dietary consumption data, and the estimation of dietary exposure is obtained through statistical processing, which determines the actual or expected amount of human exposure hazard factors. In this study, the US Environmental Protection Agency ( EPA ) point assessment model was used to calculate the acute and chronic exposure levels, respectively. Point assessment is simple and often used to screen and identify high-risk contaminants in food. The principle is to protect most of the population, which is considered to be the most suitable method for exposure screening in exposure risk assessment. The chronic health risk index HI is used to evaluate the cumulative effect of pesticides. At present, the above methods have been widely used for the health risk assessment of pesticide residues and heavy metals. Such as the follows:

1.Wu, P.L.; Wang, P.S.; Gu M.Y., Xue, J.; Wu, X. Human health risk assessment of pesticide residues in honeysuckle samples from different planting bases in China. Sci. Total. Environ. 2020, pp.1-38. doi: 10.1016/j.scitotenv.2020.142747.

2.Lemos, J.; Sampedro, M.C.; Arino, de.A.; Ortiz, A.; Barrio, R.J. Risk assessment of exposure to pesticides through dietary intake of vegetables typical of the Mediterranean diet in the Basque Country. J. Food Compos. Anal. 2016, 49, pp.35-41, 10.1016/j.jfca.2016.03.006.

3.Oliva, J.; Cermeño, S.; Cámara, M.A.; Martínez, G.; Barba, A. Disappearance of six pesticides in fresh and processed zucchini, bioavailability and health risk assessment. Food. Chem. 2017, 229, pp.172-177, doi:10.1016/j.foodchem.2017.02.076.

4.Camara, M.A.; Barba, A.; Cermeño, S.; Martinez, G.; Oliva, J. Effect of processing on the disappearance of pesticide residues in fresh-cut lettuce: Bioavailability and dietary risk. J. Environ. Sci. Health. B. 2017, 52, pp.880-886, doi:10.1080/03601234.2017.1361767.

5.Xiao, J.J.; Duan, J.S.; Xu, X.; Li, S.N.; Wang, F.; Fang, Q.K.; Liao, M.; Cao, H.Q. Behavior of pesticides and their metabolites in traditional Chinese medicine Paeoniae Radix Alba during processing and associated health risk. J. Pharm. Biomed. Anal. 2018, 161, pp.20-27, doi:10.1016/j.jpba.2018.08.029.

6.He, H.R.; Gao, F.; Zhang, Y.H.; Du P.Q.; Feng W.S.; Zheng, X.K. Effect of processing on the reduction of pesticide residues in a traditional Chinese medicine (TCM). Food. Addit. Contam. A. Chem. Anal. Control. Expo. Risk. Assess. 2020, 37, pp.1156-1164, doi:10.1080/19440049.2020.1748725.

7.Bhandari, G.; Zomer, P.; Atreya, K.; Mol, H.G.J.; Yang, X.; Geissen, V. Pesticide residues in Nepalese vegetables and potential health risks. Environ. Res. 2019, 172, pp.511-521, doi:10.1016/j.envres.2019.03.002.

8.Jankowska, M.; Kaczynski, P.; Hrynko, I.; Lozowicka, B. Dissipation of six fungicides in greenhouse-grown tomatoes with processing and health risk. Environ. Sci. Pollut. Res. Int. 2016, 23, pp.11885-11900, doi:10.1007/s11356-016-6260-x.

9.Chang, J.W.; Chen, C.Y.; Yan, B.R.; Chang, M.H.; Tseng, S.H.; Kao, Y.M.; Chen, J.C.; Lee, C.C. Cumulative risk assessment for plasticizer-contaminated food using the hazard index approach. Environ. Pollut. 2014, 189, pp.77-84, doi:10.1016/j.envpol.2014.02.005.

Point 2: Another main concern is with the article presentation. Use of the English language is not poor but the manuscript does contain many grammatical errors and some rather odd phraseology and word choice. For example at line 75 the text states ‘main grass harm’. I do not quite understand what this means. Grass is not being harmed. Grass  does not contaminate the crop. Similarly at line 60 the text states ‘cultivated artificially’ - what does this mean? I would strongly suggest the manuscript is proof read and corrected by a native English speaker before resubmission. Or sent to a professional editing service.

Response 2: The “main grass harm” has been replaced with “weeds” at line 54.

The “cultivated artificially” has been replaced with “planted artificially” at line 52.

We are sending it to a professional editing service.

Point 3: In addition the structure of the manuscript is quite poor. There is a great deal of information in the Results section which is not results but Methodology.  Paragraphing in the introduction needs a great deal of attention. The article is also far too long. There is a considerable amount of superfluous information throughout the manuscript which could easily be removed without loosing vital information. This would make the article more readable. For example, a little background is interesting in the first paragraph or two but should be removed elsewhere. Much of the background and general chat about CX in the introduction is not necessary. Elsewhere in the manuscript information given in Section 1 is repeated. Please avoid repetition. At line 298 the sentence repeats information elsewhere.

Response 3: According to the reviewer's opinion, we made significant changes to the introduction. Duplicates in the manuscript have been deleted.

Point 4: Individuals should not be named in the text e.g Line 305. Indeed most of the important information in section 3.1 could be presented in a table. Pesticide analysis information could easily be cut down. Figure 4 adds very little value.

A short summary of the methodological approach should be included in the Abstract.

The methodology section needs more work. Method description in the results section needs to repositioned. Approaches needed to be justified.

Response 4: The purpose of this paper is to analyze pesticide residues and risk assessment of Chuanxiong Rhizoma. The detection methods of pesticide residues in Chuanxiong Rhizoma were GC-MS/MS and UPLC-MS/MS methods established in the early stage of the research group. According to the opinions of the reviewers, precision/RSD (%) was added in Table 1, and tables 4 and 5 were added at the same time. Figures 3 and 4 have been enlarged and blank, recovery, and sample TIC diagrams have been added.

Point 5: Please take care to ensure that text in the Results section does not simply repeat what is obvious from the data in tables. Important insights only should be repeated and these should add value to the tables.

Response 5: We have added references, and compared the research results with the reported contents.

Point 6: Referencing is also inadequate in places. There are many stated facts that are not referenced. For example at line 152 a reference is needed related to reduced degradation half-lives and references are needed to support the manuals and literature mentioned at line 224. There are several other examples in the text.

Response 6: Lines 149, 172, and 224 have supplemented the references.

Point 7: Other comments

 - All abbreviations in the main text should be spelt out the first time they are used. Abbreviations should be avoided in the abstract. SFDA needs to spelt out in full at line 50 as well as TCM at line 60.

- hyperlinks in the text should be converted to a reference.

- At line 74 ‘red spiders’ should read ‘red spider mite’. These insects are not arachnids.

- why was dichlorvos mentioned at line 104 rather than other pesticides?

Response 7: SFDA has been removed.

TCM has been spelled out at line 48.

The “red spiders” have been replaced with “red spider mite” at line 55.

The dichlorvos has been removed at line104.

Link to Chinese references of Chuanxiong Rhizoma: https://sicaumaize-my.sharepoint.com/:f:/g/personal/zhangdelin_sicaumaize_onmicrosoft_com/ElINk1V8rBVLlnfovfvH8i4Brl8MoSoPLal992amDbtxNg?e=lOCZ2P

Reviewer 3 Report

The manuscript “Detection and risk assessments of multi-pesticides in traditional Chinese medicine Chuanxiong Rhizoma by LC/MS-MS and GC/MS-MS” concerns the determination of several pesticides in a large number of traditional Chinese herb samples used in medicine and food and their long-term, short-term and cumulative risk assessment. The topic of this work relates to human exposure to pesticides through the use of Chuanxiong Rhizoma, and it is of significant interest since it concerns human health and safety. However, there are significant deficiencies of this study:

1) The novelty and the purpose of this study is not clearly reported and justified.

2) The Introduction presents a lot of information of local interest and limited information of the scientific data available on the topic of the manuscript like the analytical methods that have been applied for these kinds of determinations, and the relevant risk assessments. The authors claim that there are a few studies related to pesticides residues in CX in Lines 94-99. However, this is not fully justified based on the following available publications:

- Rapid Screening and Quantitative Analysis of 74 Pesticide Residues in Herb by Retention Index Combined with GC-QQQ-MS/MS, 2021, Journal of Analytical Methods in Chemistry 2021,8816854

- Comparison of two extraction methods for the determination of 135 pesticides in Corydalis Rhizoma, Chuanxiong Rhizoma and Angelicae Sinensis Radix by liquid chromatography-triple quadrupole-mass spectrometry. Application to the roots and rhizomes of Chinese herbal medicines, 2016, Journal of Chromatography B: Analytical Technologies in the Biomedical and Life Sciences 1017-1018, pp. 233-240

- A multi-residue method for simultaneous determination of 74 pesticides in Chinese material medica using modified QuEChERS sample preparation procedure and gas chromatography tandem mass spectrometry, 2016, Journal of Chromatography B: Analytical Technologies in the Biomedical and Life Sciences, 1015-1016, pp. 1-12

It is noted that the above articles from the international literature are not included by the authors in the manuscript. Furthermore, the cited references [13 -16] either they could not be tracked or they were written in Chinese.

3) The authors should compare their findings with the corresponding findings of the literature.

4) The analytical methods applied (LC-MS/MS and GC-MS/MS) are not described adequately. For example, the MS/MS transitions of each analyte should be reported (the m/z values). One or more transitions were selected for the determination of each analyte? The validation concerns one transition? Although there is reference to other publications [13] and [19] it is noted that reference [13] could not be tracked.  In addition, the description of the validation of the methods is very limited. How were the recovery experiments conducted? The recovery values and %RSD correspond to which concentration levels?

- The concentrations of the calibration curve is given in ng/ml in Table 1. To which concentrations of sample expressed as μg/kg doe they correspond to? The number of the calibration levels and the number of their replicates should be added

- Too many significant figures in the linear equation and concentrations. Not presented in a consistent way. 

- The chromatograms in Figures 3 and 4 are not legible. The peaks should be correlated to the compounds and chromatograms of the real samples should be presented.

5) Table 2:

Title of column 3 (Serviceable range) should be rephrased

Title of column 4 (Detected batch) should be rephrased to “Number of batches Detected”

 Does the MRL refer to the specific herb?

6) abbreviations should be explained at first appearance in the text (ex: GAP, IEDI, ADI, ARfD, JMPR…)

7) Lines 79-80: The authors report thiophanate-methyl as conventional and as banned pesticide at the same time. They should check the information and make clearer in the text for which countries thiophanate-methyl and the other “banned” pesticides are not approved? the relevant Regulations should be cited.

8) Line 207-208: the MRL of chlorpyrifos 0.2 mg/kg refers to which matrix exactly?

9) Line 291 – 292: What is the criteria based on which it is concluded that the cumulative intake of pesticides in CX will not cause health damage? An appropriate reference should be cited.

10) Lines 302 – 304: The samples were collected from the field and were obtained from the local farmers or were they commercially available products obtained from the market?

11) Lines 309-313: Use of English requires improvement

12) Line 313: How is the “high toxicity” justified?

13) Table 4 should be removed from the main text since its content is not essential but just informative. It could be added to a supplementary material.

14) Line 337: What does NaAC stand for?

15) Line 347: Which compound was used as internal standard for the GC-MS analysis. This information should be also added in 3.2 Reagents and Materials

16) Lines 430-441: Probably “CR” should be replaced with “CX”.

17) The use of English requires significant improvement

Author Response

Dear editors and references, thank you for your very careful review and constructive comments on our manuscript (manuscript No.: molecules-1467711) " Detection and risk assessments of multi-pesticides in traditional Chinese medicine Chuanxiong Rhizoma by LC/MS-MS and GC/MS-MS ". Taking into account all the corrections and suggestions, the manuscript has been revised. We are very grateful to the reviewers for pointing out that our manuscript needs to be improved. The points raised in the reply are as follows:

Point 1: The novelty and the purpose of this study is not clearly reported and justified.

Response 1: According to the reviewer's opinion, we made significant changes to the introduction. Compared with previous studies, our samples had the characteristics of more collection sites, a larger sample size, and stronger representativeness. At the same time, the characteristics of GC-MS/MS and UPLC-MS/MS were used to detect a variety of pesticide residues and carry out the health risk assessment, while the health risk assessment of Chuanxiong Rhizoma has not been reported.

Point 2: The Introduction presents a lot of information of local interest and limited information of the scientific data available on the topic of the manuscript like the analytical methods that have been applied for these kinds of determinations, and the relevant risk assessments. The authors claim that there are a few studies related to pesticides residues in CX in Lines 94-99. However, this is not fully justified based on the following available publications:

- Rapid Screening and Quantitative Analysis of 74 Pesticide Residues in Herb by Retention Index Combined with GC-QQQ-MS/MS, 2021, Journal of Analytical Methods in Chemistry 2021,8816854

- Comparison of two extraction methods for the determination of 135 pesticides in Corydalis Rhizoma, Chuanxiong Rhizoma and Angelicae Sinensis Radix by liquid chromatography-triple quadrupole-mass spectrometry. Application to the roots and rhizomes of Chinese herbal medicines, 2016, Journal of Chromatography B: Analytical Technologies in the Biomedical and Life Sciences 1017-1018, pp. 233-240

- A multi-residue method for simultaneous determination of 74 pesticides in Chinese material medica using modified QuEChERS sample preparation procedure and gas chromatography tandem mass spectrometry, 2016, Journal of Chromatography B: Analytical Technologies in the Biomedical and Life Sciences, 1015-1016, pp. 1-12

It is noted that the above articles from the international literature are not included by the authors in the manuscript. Furthermore, the cited references [13 -16] either they could not be tracked or they were written in Chinese.

Response 2: According to the reviewer's opinion, we made significant changes to the introduction. The above three articles related to pesticide residues of Chuanxiong Rhizoma were included in the manuscript. Chuanxiong Rhizoma is one of the Chinese herbal medicines. At present, the reports on pesticide residues in Chuanxiong Rhizoma are mainly in Chinese, and the above four articles are all in Chinese. We will upload it as an attachment for reviewers.

Point 3: The authors should compare their findings with the corresponding findings of the literature.

Response 3: According to the reviewer's opinion, we added and compared the published pesticide residue detection results in document “ 2.2.Pesticide Residues Concentrations”.

Point 4: The analytical methods applied (LC-MS/MS and GC-MS/MS) are not described adequately. For example, the MS/MS transitions of each analyte should be reported (the m/z values). One or more transitions were selected for the determination of each analyte? The validation concerns one transition? Although there is reference to other publications [13] and [19] it is noted that reference [13] could not be tracked.  In addition, the description of the validation of the methods is very limited. How were the recovery experiments conducted? The recovery values and %RSD correspond to which concentration levels?

- The concentrations of the calibration curve is given in ng/ml in Table 1. To which concentrations of sample expressed as μg/kg doe they correspond to? The number of the calibration levels and the number of their replicates should be added

- Too many significant figures in the linear equation and concentrations. Not presented consistently.

- The chromatograms in Figures 3 and 4 are not legible. The peaks should be correlated to the compounds and chromatograms of the real samples should be presented.

Response 4: The purpose of this paper is to analyze pesticide residues and risk assessment of Chuanxiong Rhizoma. The detection methods of pesticide residues in Chuanxiong Rhizoma were GC-MS/MS and UPLC-MS/MS methods established in the early stage of the research group. Because of the space problem, the verification method, recovery rate, and precision were not described in detail. The author of reference [13] will upload it as an attachment for reviewers to review. According to the opinions of the reviewers, precision/RSD (%) was added in Table 1, and tables 4 and 5 were added at the same time. Significant figures in the linear equation and concentrations were presented consistently. Figures 3 and 4 have been enlarged and blank, recovery, and sample TIC diagrams have been added. Due to a large number of compounds and information, there is no labeling here.

Point 5: Table 2:

Title of column 3 (Serviceable range) should be rephrased

Title of column 4 (Detected batch) should be rephrased to “Number of batches Detected”

 Does the MRL refer to the specific herb?

Response 5: Columns 3 and 4 of Table 2 have been revised following the opinions of the reviewer. MRL is widely used in herbal medicines, vegetables, fruits, water and processed foods.

Point 6: abbreviations should be explained at first appearance in the text (ex: GAP, IEDI, ADI, ARfD, JMPR…)

Response 6: Revise the manuscript in the corresponding position according to the opinions of the reviewers.

Point 7: Lines 79-80: The authors report thiophanate-methyl as conventional and as banned pesticide at the same time. They should check the information and make clearer in the text for which countries thiophanate-methyl and the other “banned” pesticides are not approved? the relevant Regulations should be cited.

Response 7: Here is our error, and thiophanate-methyl is a conventional pesticide, which is not included in the banned pesticide list.

Point 8: Line 207-208: the MRL of chlorpyrifos 0.2 mg/kg refers to which matrix exactly?

Response 8: The original European Pharmacopoeia 9.0 p:286-287 was “Unless otherwise indicated in the monograph, the herbal drug to be examined at least conforms to the limit indicated in Table 2.8.13.-1”, Table 2.8.13.-1 list chlorpyrifos MRL 0.2 mg/kg. Therefore, the MRL of 0.2 mg/kg was used as the MRL of chlorpyrifos in Chuanxiong for comparative analysis.

Point 9: Line 291 – 292: What is the criteria based on which it is concluded that the cumulative intake of pesticides in CX will not cause health damage? An appropriate reference should be cited.

Response 9: When HI > 1, the CX involved should be considered a risk to consumers, but if HI is < 1, the CX is considered acceptable, detailed description under 3.4.3. Cumulative Risk Assessment. We cited 3 appropriate references at Line 290.

Point 10: Lines 302 – 304: The samples were collected from the field and were obtained from the local farmers or were they commercially available products obtained from the market?

Response 10: The samples were collected from the field and were obtained from the local farmers. This information has been detailed in “3.1. Sample Collection”.

Point 11: Lines 309-313: Use of English requires improvement

Response 11: We have removed line lines 309-313.

Point 12: Line 313: How is the “high toxicity” justified?

Response 12: We have removed line 313.

Point 13: Table 4 should be removed from the main text since its content is not essential but just informative. It could be added to a supplementary material.

Response 13: Table 4 has been removed.

Point 14:  Line 337: What does NaAC stand for?

Response 14: NaAC is sodium acetate, and it has been marked on line 305.

Point 15:  Line 347: Which compound was used as internal standard for the GC-MS analysis. This information should be also added in 3.2 Reagents and Materials

Response 15: Triphenyl phosphate was used as the internal standard for the GC-MS analysis. This information has been added in “3.2 Reagents and Materials”.

Point 16: Lines 430-441: Probably “CR” should be replaced with “CX”.

Response 16: According to the reviewer's opinion, "CR" has been replaced with "CX" in lines 430-441.

Point 17:  The use of English requires significant improvement.

Response 17: The English language was modified according to the opinions of reviewers.

Link to Chinese references of Chuanxiong Rhizoma: https://sicaumaize-my.sharepoint.com/:f:/g/personal/zhangdelin_sicaumaize_onmicrosoft_com/ElINk1V8rBVLlnfovfvH8i4Brl8MoSoPLal992amDbtxNg?e=lOCZ2P

Round 2

Reviewer 2 Report

This is an interesting manuscript that describes a study that seeks to consider the levels and thus exposure and risk from pesticide residues present in a common herb, Chuanxiong Rhizoma (CX), used in traditional Chinese medicine. The manuscript topic does fall within the scope of the journal but I believe it would gain much more exposure in a journal more aligned with public health.

This is the second time I have reviewed this manuscript. Following revision many corrections have been made and it now reads somewhat better. As a risk assessor I still feel that the method is out of date but thats a professional opinion. I am sure readers will make their own mind up.

Author Response

Dear editors and references, thank you for your very careful review and constructive comments on our manuscript (manuscript No.: molecules-1467711) " Detection and risk assessments of multi-pesticides in traditional Chinese medicine Chuanxiong Rhizoma by LC/MS-MS and GC/MS-MS ". Taking into account all the corrections and suggestions, the manuscript has been revised. We are very grateful to the reviewers for pointing out that our manuscript needs to be improved. The points raised in the reply are as follows:

Point 1: As a risk assessor I still feel that the method is out of date but thats a professional opinion. I am sure readers will make their own mind up.

Response 1: According to the reviewer's opinion, we modified the EDI and ESTI formulae to include the exposed days over a lifetime, exposure frequency, and average lifetime reported in the literature for Chinese people ingesting traditional Chinese medicine.

Reviewer 3 Report

Overall the authors have improved significantly their manuscript addressing most of the reviewer's comments. There are still some points that need to be further elaborated. 

Point 1: The novelty and the purpose of this study is not clearly reported and justified.

Response 1: According to the reviewer's opinion, we made significant changes to the introduction. Compared with previous studies, our samples had the characteristics of more collection sites, a larger sample size, and stronger representativeness. At the same time, the characteristics of GC-MS/MS and UPLC-MS/MS were used to detect a variety of pesticide residues and carry out the health risk assessment, while the health risk assessment of Chuanxiong Rhizoma has not been reported.

Reviewer response 1: The authors improved the Introduction making the purpose of the study clearer, however the novelty of their work is still not reported in the manuscript. Their argumentation presented in their Response 1 should be transferred in the actual manuscript as well, and in particular in the Introduction, for the benefit of the reader.

Point 4: The analytical methods applied (LC-MS/MS and GC-MS/MS) are not described adequately. For example, the MS/MS transitions of each analyte should be reported (the m/z values). One or more transitions were selected for the determination of each analyte? The validation concerns one transition? Although there is reference to other publications [13] and [19] it is noted that reference [13] could not be tracked.  In addition, the description of the validation of the methods is very limited. How were the recovery experiments conducted? The recovery values and %RSD correspond to which concentration levels?

- The concentrations of the calibration curve is given in ng/ml in Table 1. To which concentrations of sample expressed as μg/kg doe they correspond to? The number of the calibration levels and the number of their replicates should be added

- Too many significant figures in the linear equation and concentrations. Not presented consistently.

- The chromatograms in Figures 3 and 4 are not legible. The peaks should be correlated to the compounds and chromatograms of the real samples should be presented.

Response 4: The purpose of this paper is to analyze pesticide residues and risk assessment of Chuanxiong Rhizoma. The detection methods of pesticide residues in Chuanxiong Rhizoma were GC-MS/MS and UPLC-MS/MS methods established in the early stage of the research group. Because of the space problem, the verification method, recovery rate, and precision were not described in detail. The author of reference [13] will upload it as an attachment for reviewers to review. According to the opinions of the reviewers, precision/RSD (%) was added in Table 1, and tables 4 and 5 were added at the same time. Significant figures in the linear equation and concentrations were presented consistently. Figures 3 and 4 have been enlarged and blank, recovery, and sample TIC diagrams have been added. Due to a large number of compounds and information, there is no labeling here.

 Reviewer response 4: Authors improved the presentation of the methods, however they removed the reference to their previous work. They should add it again since the description of the validation of the methods is  very limited and cannot stand alone without relative references. The quality of the chromatograms is not improved, authors should try to provide Figures of better quality.  

Reviewer response 5: Please replace “Forbid” with “Forbidden” in Table 2

Point 7: Lines 79-80: The authors report thiophanate-methyl as conventional and as banned pesticide at the same time. They should check the information and make clearer in the text for which countries thiophanate-methyl and the other “banned” pesticides are not approved? the relevant Regulations should be cited.

Response 7: Here is our error, and thiophanate-methyl is a conventional pesticide, which is not included in the banned pesticide list.

 Reviewer response 7: The authors’ response is not in full agreement with the manuscript since in “Conclusions” they report that thiophanate methyl is banned. They should revise again and clarify if they refer to China. For example thiophanate-methyl is currently not approved in Europe.

Further considerations on the use of English

Line 18: “actuality”?

Line 91: “to detect a strong polar” ?

Line 94: “CX, We investigated”?

Lines 101- 103: Syntax error

Line 126: “guideless” probably the authors mean “guidelines”. A relative reference should be added as well for these guidelines to be more precise.

Author Response

Dear editors and references, thank you for your very careful review and constructive comments on our manuscript (manuscript No.: molecules-1467711) " Detection and risk assessments of multi-pesticides in traditional Chinese medicine Chuanxiong Rhizoma by LC/MS-MS and GC/MS-MS ". Taking into account all the corrections and suggestions, the manuscript has been revised. We are very grateful to the reviewers for pointing out that our manuscript needs to be improved. The points raised in the reply are as follows:

Point 1: The authors improved the Introduction making the purpose of the study clearer, however the novelty of their work is still not reported in the manuscript. Their argumentation presented in their Response 1 should be transferred in the actual manuscript as well, and in particular in the Introduction, for the benefit of the reader.

Response 1: According to the reviewer's opinion, We modify lines 74-80 and 119-123 to highlight the novelty of manuscripts.

Point 2: Authors improved the presentation of the methods, however they removed the reference to their previous work. They should add it again since the description of the validation of the methods is  very limited and cannot stand alone without relative references. The quality of the chromatograms is not improved, authors should try to provide Figures of better quality. 

Response 2: We have added references to previous work at lines 314-318. About the chromatograms, the reviewer gave us a good suggestion, and we would like to put higher quality chromatograms into our manuscript, but the chromatogram was already better quality than many chromatograms. So it's hard for us to do this.

Point 3: Please replace “Forbid” with “Forbidden” in Table 2.

Response 3: According to the reviewer's opinion, we have replaced “Forbid” with “Forbidden” in table 2.

Point 4: The authors’ response is not in full agreement with the manuscript since in “Conclusions” they report that thiophanate methyl is banned. They should revise again and clarify if they refer to China. For example thiophanate-methyl is currently not approved in Europe.

Response 4: We checked China and other national standards and found that thiophanate-methyl was not listed as a banned pesticide. Only Canada considered the impact on human health and suggested canceling multiple uses.

Point 5: Line 18: “actuality”?

Line 91: “to detect a strong polar” ?

Line 94: “CX, We investigated”?

Lines 101- 103: Syntax error

Line 126: “guideless” probably the authors mean “guidelines”. A relative reference should be added as well for these guidelines to be more precise.

Response 5: Line 18: we have replaced “actuality” with “status”.

Line 91: We deleted “a”.

Line 94: We modified lines 94-96 about the description of the investigation on Chuanxiong.

Lines 101- 103: We modified lines 105-107.

Line 126:we have replaced “guideless” with “guidelines”.
